# Universal Semantic Parsing

## Abstract

Universal Dependencies (UD) provides a cross-linguistically uniform syntactic representation, with the aim of advancing multilingual applications of parsing and natural language understanding. Reddy et al. (2016) recently developed a semantic interface for (English) Stanford Dependencies, based on the lambda calculus. In this work, we introduce UDEPLAMBDA, a similar semantic interface for UD, which allows mapping natural language to logical forms in an almost language-independent framework. We evaluate our approach on semantic parsing for the task of question answering against Freebase. To facilitate multilingual evaluation, we provide German and Spanish translations of the WebQuestions and GraphQuestions datasets. Results show that UDEPLAMBDA outperforms strong baselines across languages and datasets. For English, it achieves the strongest result to date on GraphQuestions, with competitive results on WebQuestions.

## 1 Introduction

The Universal Dependencies (UD) initiative seeks to develop cross-linguistically consistent annotation guidelines as well as a large number of uniformly annotated treebanks for many languages.[1] Such resources could advance multilingual applications of parsing, improve comparability of evaluation results, enable cross-lingual learning, and more generally support natural language understanding.

Seeking to exploit the benefits of UD for natural language understanding, we introduce UDEP-LAMBDA, a semantic interface for UD that maps natural language to logical forms, representing underlying predicate-argument structures, in an almost language-independent manner. Our framework is based on DEPLAMBDA (Reddy et al., 2016) a recently developed method that converts English Stanford Dependencies (SD) to logical forms. The conversion process is illustrated in Figure 1 and discussed in more detail in Section 2. Whereas DEP-LAMBDA works only for English, UDEPLAMBDA applies to any language for which UD annotations are available.[2] In this paper, we describe the rationale behind UDEPLAMBDA and highlight important differences from DEPLAMBDA, some of which stem from the different treatment of various linguistic constructions in UD.

Our experiments focus on semantic parsing as a testbed for evaluating the framework's multilingual appeal. We address the task of learning to map natural language to machine interpretable formal meaning representations, specifically retrieving answers to questions from Freebase. To facilitate multilingual evaluation, we provide translations of the English WebQuestions (Berant et al., 2013) and GraphQuestions (Su et al., 2016) datasets to German and Spanish. We demonstrate that U-DEPLAMBDA can be used to derive logical forms for these languages using a minimal amount of language-specific knowledge. Aside from developing the first multilingual semantic parsing tool for Freebase, we also experimentally show that U-DEPLAMBDA outperforms strong baselines across languages and datasets. For English, it achieves the strongest result to date on GraphQuestions, with competitive results on WebQuestions. Beyond semantic parsing, we believe that the logical forms produced by our framework will be of use in various natural understanding tasks including entailment (Beltagy et al., 2016), text-based question

---

[1] http://www.universaldependencies.org/.

[2] As of v1.3, UD annotations are available for 47 languages.

answering (Lewis and Steedman, 2013), sentence simplification (Narayan and Gardent, 2014), summarization (Liu et al., 2015), paraphrasing (Pavlick et al., 2015), and relation extraction (Rocktäschel et al., 2015). Our implementation and translated datasets will be made publicly available.

## 2 DEPLAMBDA

Before describing UDEPLAMBDA, we provide an overview of DEPLAMBDA (Reddy et al., 2016) on which our approach is based. DEPLAMBDA converts a dependency tree to its logical form in three steps: *binarization*, *substitution*, and *composition*, each of which is briefly outlined below.

**Binarization** A dependency tree is first mapped to a Lisp-style s-expression indicating the order of semantic composition. Figure 1(b) shows the s-expression for the sentence *Disney won an Oscar for the movie Frozen*, derived from the dependency tree in Figure 1(a). Here, the sub-expression (dobj won (det Oscar an)) indicates that the logical form of the phrase *won an Oscar* is derived by composing the logical form of the label dobj with the logical form of the word *won* and the logical form of the phrase *an Oscar*, derived analogously.

An *obliqueness hierarchy* is employed to impose a strict ordering on the modifiers to each head in the dependency tree. As an example, *won* has three modifiers in Figure 1(a), which according to the obliqueness hierarchy are composed in the order dobj > nmod > nsubj. In constructions like coordination, this ordering is crucial to arrive at the correct semantics (see Section 3.3).

**Substitution** Each symbol in the s-expressions is substituted for a lambda expression encoding its semantics. Words and dependency labels are assigned different types of expressions. In general, words have expressions of the following kind:

ENTITY $\Rightarrow \lambda x.\text{word}(x_a)$; e.g. Oscar $\Rightarrow \lambda x.\text{Oscar}(x_a)$
EVENT $\Rightarrow \lambda x.\text{word}(x_e)$; e.g. won $\Rightarrow \lambda x.\text{won}(x_e)$
FUNCTIONAL $\Rightarrow \lambda x.\text{TRUE}$; e.g. an $\Rightarrow \lambda x.\text{TRUE}$

Here, the subscripts $\cdot_a$ and $\cdot_e$ denote the types of individuals (**Ind**) and events (**Event**), respectively, whereas $x$ denotes a paired variable $(x_a, x_e)$ of type **Ind** × **Event**. Roughly speaking, proper nouns and adjectives invoke ENTITY expressions, verbs and adverbs invoke EVENT expressions, and common nouns invoke both ENTITY and EVENT expressions (see Section 3.3), while remaining words invoke FUNCTIONAL expressions. As in DEPLAMBDA,

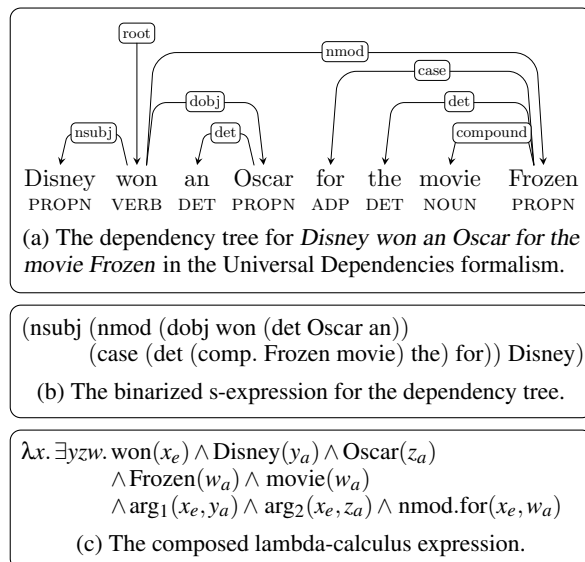

(a) The dependency tree for *Disney won an Oscar for the movie Frozen* in the Universal Dependencies formalism.

(nsubj (nmod (dobj won (det Oscar an))
(case (det (comp. Frozen movie) the) for)) Disney)

(b) The binarized s-expression for the dependency tree.

$\lambda x. \exists yzw. \text{won}(x_e) \wedge \text{Disney}(y_a) \wedge \text{Oscar}(z_a)$
$\wedge \text{Frozen}(w_a) \wedge \text{movie}(w_a)$
$\wedge \text{arg}_1(x_e, y_a) \wedge \text{arg}_2(x_e, z_a) \wedge \text{nmod.for}(x_e, w_a)$

(c) The composed lambda-calculus expression.

Figure 1: The mapping of a dependency tree to its logical form with the intermediate s-expression.

we enforce the constraint that every s-expression is of the type $\eta = \textbf{Ind} \times \textbf{Event} \rightarrow \textbf{Bool}$, which simplifies the type system considerably.

Expressions for dependency labels glue the semantics of heads and modifiers to articulate predicate-argument structure. These expressions in general take one of the following forms:

COPY $\Rightarrow \lambda fgx. \exists y. f(x) \wedge g(y) \wedge \text{rel}(x,y)$
e.g. nsubj, dobj, nmod, advmod
INVERT $\Rightarrow \lambda fgx. \exists y. f(x) \wedge g(y) \wedge \text{rel}^i(y,x)$
e.g. amod, acl
MERGE $\Rightarrow \lambda fgx. f(x) \wedge g(x)$
e.g. compound, appos, amod, acl
HEAD $\Rightarrow \lambda fgx. f(x)$
e.g. case, punct, aux, mark.

As an example of COPY, consider the lambda expression for dobj in (dobj won (det Oscar an)): $\lambda fgx. \exists y. f(x) \wedge g(y) \wedge \text{arg}_2(x_e, y_a)$. This expression takes two functions $f$ and $g$ as input, where $f$ represents the logical form of *won* and $g$ represents the logical form of *an Oscar*. The predicate-argument structure $\text{arg}_2(x_e, y_a)$ indicates that the $\text{arg}_2$ of the event $x_e$, i.e. *won*, is the individual $y_a$, i.e. the entity *Oscar*. Since $\text{arg}_2(x_e, y_a)$ mimics the dependency structure dobj(won, Oscar), we refer to the expression kind evoked by dobj as COPY.

Expressions that invert the dependency direction are referred to as INVERT (e.g. amod in *running horse*); expressions that merge two subexpressions without introducing any relation predicates are referred to as MERGE (e.g. compound in *movie Frozen*); and expressions that simply return the parent expression semantics are referred to as HEAD

(e.g. case in *for Frozen*). While this generalization applies to most dependency labels, several labels take a different logical form not listed here, some of which are discussed in Section 3.3. Sometimes the mapping of dependency label to lambda expression may depend on surrounding part-of-speech tags or dependency labels. For example, amod acts as INVERT when the modifier is a verb (e.g. in *running horse*), and as MERGE when the modifier is an adjective (e.g. in *beautiful horse*).[3]

**Composition**   The final logical form is computed by beta-reduction, treating expressions of the form (f x y) as the function f applied to the arguments x and y. For example, (dobj won (det Oscar an)) results in $\lambda x. \exists z. \text{won}(x_e) \wedge \text{Oscar}(z_a) \wedge \text{arg}_2(x_e, z_a)$ when the expression for dobj is applied to those for *won* and *(det Oscar an)*. Figure 1(c) shows the logical form for the s-expression in Figure 1(b).

## 3   UDEPLAMBDA

We now introduce UDEPLAMBDA, a semantic interface for Universal Dependencies.[4]   Whereas DEPLAMBDA only applies to English Stanford Dependencies, UDEPLAMBDA takes advantage of the cross-lingual nature of UD to facilitate an (almost) language independent semantic interface. This is accomplished by restricting the binarization, substitution, and composition steps described above to rely solely on information encoded in the UD representation. Importantly, UDEPLAMBDA is designed to not rely on lexical forms in a language to assign lambda expressions, but only on information contained in dependency labels and postags.

However, some linguistic phenomena are language specific (e.g. pronoun-dropping) or meaning specific (e.g. *every* and *the* in English have very different semantics, despite being both determiners) and are not encoded in the UD schema. Furthermore, some cross-linguistic phenomena, such as long-distance dependencies, are not part of the core UD representation. To circumvent this limitation, a simple *enhancement* step enriches the original UD representation before binarization takes place (Section 3.1). This step adds to the dependency tree missing syntactic information and long-distance dependencies, thereby creating a

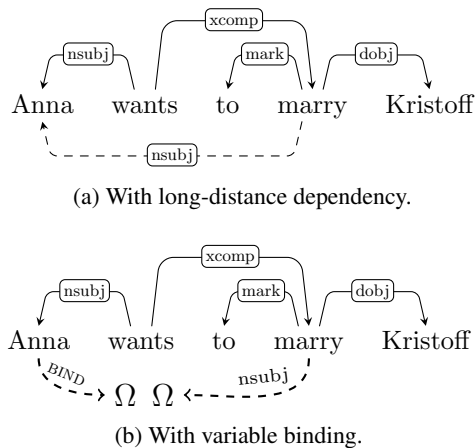

(a) With long-distance dependency.

(b) With variable binding.

Figure 2: The original and enhanced dependency trees for *Anna wants to marry Kristoff*.

graph. Whereas DEPLAMBDA is not able to handle graph-structured input, UDEPLAMBDA is designed to work directly with dependency graphs (Section 3.2). Finally, the representation of several linguistic constructions differ between UD and SD, which requires different handling in the semantic interface (Section 3.3).

### 3.1   Enhancement

Both Schuster and Manning (2016) and Nivre et al. (2016) note the necessity of an enhanced UD representation to enable semantic applications. However, such enhancements are currently only available for a subset of languages in UD. Instead, we rely on a small number of enhancements for our main application—semantic parsing for question-answering—with the hope that this step can be replaced by an enhanced UD representation in the future. Specifically, we define three kinds of enhancements: (1) long-distance dependencies; (2) types of coordination; and (3) refined question word tags.

First, we identify long-distance dependencies in relative clauses and control constructions. We follow Schuster and Manning (2016) and find these using the labels acl (relative) and xcomp (control). Figure 2(a) shows the long-distance dependency in the sentence *Anna wants to marry Kristoff*. Here, *marry* is provided with its missing nsubj (dashed arc). Second, UD conflates all coordinating constructions to a single dependency label, conj. To obtain the correct coordination scope, we refine conj to conj:verb, conj:vp, conj:sentence, conj:np, and conj:adj, similar to Reddy et al. (2016). Finally, unlike the PTB tags (Marcus et al., 1993) used by SD, the UD part-of-speech tags do

---

[3]We use Tregex (Levy and Andrew, 2006) for substitution mappings and Cornell SPF (Artzi, 2013) as the lambda-calculus implementation. For example, in *running horse*, the tregex */label:amod/=target < /postag:verb/* matches amod to its INVERT expression $\lambda f g x. \exists y. f(x) \wedge g(y) \wedge \text{amod}^i(y_e, x_a)$.

[4]In what follows, all references to UD are to UD v1.3.

not distinguish question words. Since these are crucial to question-answering, we use a small lexicon to refine the tags for determiners (DET), adverbs (ADV) and pronouns (PRON) to DET:WH, ADV:WH and PRON:WH, respectively. Specifically, we use a list of 12 (English), 14 (Spanish) and 35 (German) words, respectively. This is the only part of UDEPLAMBDA that relies on language-specific information. We hope that, as the coverage of morphological features in UD improves, this refinement can be replaced by relying on morphological features, such as the interrogative feature (INT).

### 3.2 Graph Structures and BIND

To handle graph structures that may result from the enhancement step, such as those in Figure 2(a), we propose a variable-binding mechanism that differs from that of DEPLAMBDA. First, each long-distance dependency is split into independent arcs as shown in Figure 2(b). Here, $\Omega$ is a placeholder for the subject of *marry*, which in turn corresponds to *Anna* as indicated by the binding of $\Omega$ via the pseudo-label BIND. We treat BIND like an ordinary dependency label with semantics MERGE and process the resulting tree as usual, via the s-expression:

(nsubj (xcomp wants (nsubj (mark
  (dobj marry Kristoff) to) $\Omega$) (BIND Anna $\Omega$)),

with the lambda-expression substitutions:

*wants, marry* $\in$ EVENT; *to* $\in$ FUNCTIONAL;
*Anna, Kristoff* $\in$ ENTITY;
mark $\in$ HEAD; BIND $\in$ MERGE;
xcomp $= \lambda fgx. \exists y. f(x) \wedge g(y) \wedge \text{xcomp}(x_e, y_e)$.

These substitutions are based solely on unlexicalized context. For example, the part-of-speech tag PROPN of *Anna* invokes an ENTITY expression.

The placeholder $\Omega$ has semantics $\lambda x. \text{EQ}(x, \omega)$, where $\text{EQ}(u, \omega)$ is true iff $u$ and $\omega$ are equal (have the same denotation), which unifies the subject variable of *wants* with the subject variable of *marry*.

After substitution and composition, we get:

$\lambda z. \exists xywv. \text{wants}(z_e) \wedge \text{Anna}(x_a) \wedge \text{arg}_1(z_e, x_a) \wedge \text{EQ}(x, \omega)$
$\wedge \text{marry}(y_e) \wedge \text{xcomp}(z_e, y_e) \wedge \text{arg}_1(y_e, v_a) \wedge \text{EQ}(v, \omega)$
$\wedge \text{Kristoff}(w_a) \wedge \text{arg}_2(y_e, w_a)$,

This expression may be simplified further by replacing all occurrences of $v$ with $x$ and removing the unification predicates EQ, which results in:

$\lambda z. \exists xyw. \text{wants}(z_e) \wedge \text{Anna}(x_a) \wedge \text{arg}_1(z_e, x_a)$
$\wedge \text{marry}(y_e) \wedge \text{xcomp}(z_e, y_e) \wedge \text{arg}_1(y_e, x_a)$
$\wedge \text{Kristoff}(w_a) \wedge \text{arg}_2(y_e, w_a)$.

This expression encodes the fact that *Anna* is the arg$_1$ of the *marry* event, as desired. DEPLAMBDA, in contrast, cannot handle graph-structured input,

since it lacks a principled way of generating s-expressions from graphs. Even given the above s-expression, BIND in DEPLAMBDA is defined in a way such that the composition fails to unify $v$ and $x$, which is crucial for the correct semantics. Moreover, the definition of BIND in DEPLAMBDA does not have a formal interpretation within the lambda calculus, unlike ours.

### 3.3 Linguistic Constructions

Below, we highlight the most pertinent differences between UDEPLAMBDA and DEPLAMBDA, stemming from the different treatment of various linguistic constructions in UD versus SD.

**Prepositional Phrases** UD uses a content-head analysis, in contrast to SD, which treats function words as heads of prepositional phrases, Accordingly, the s-expression for the phrase *president in 2009* is (nmod president (case 2009 in)) in UDEPLAMBDA and (prep president (pobj in 2009)) in DEPLAMBDA. To achieve the desired semantics,

$\lambda x. \exists y. \text{president}(x_a) \wedge \text{president\_event}(x_e) \wedge$
$\quad \text{arg}_1(x_e, x_a) \wedge 2009(y_a) \wedge \text{prep.in}(x_e, y_a)$,

DEPLAMBDA relies on an intermediate logical form that requires some post-processing, whereas UDEPLAMBDA obtains the desired logical form directly through the following entries:

*in* $\in$ FUNCTIONAL; *2009* $\in$ ENTITY; case $\in$ HEAD;
*president* $= \lambda x. \text{president}(x_a) \wedge \text{president\_event}(x_e)$
$\quad \wedge \text{arg}_1(x_e, x_a)$;
nmod $= \lambda fgx. \exists y. f(x) \wedge g(y) \wedge \text{nmod.in}(x_e, y_a)$.

Other nmod constructions, such as possessives (nmod:poss), temporal modifiers (nmod:tmod) and adverbial modifiers (nmod:npmod), are handled similarly. Note how the common noun *president*, evokes both entity and event predicates above.

**Passives** DEPLAMBDA gives special treatment to passive verbs, identified by the fine-grained part-of-speech tags in the PTB tag together with dependency context. For example, *An Oscar was won* is analyzed as $\lambda x. \text{won.pass}(x_e) \wedge \text{Oscar}(y_a) \wedge \text{arg}_1(x_e, y_a)$, where won.pass represents a passive event. However, UD does not distinguish between active and passive forms.[5] While the labels nsubjpass or auxpass indicate passive constructions, such clues are sometimes missing, such as in reduced relatives. We therefore opt to not have separate entries for passives, but aim to produce identical logical forms for active and passive forms when

---

[5]UD encodes voice as a morphological feature, but most syntactic analyzers do not produce this information yet.

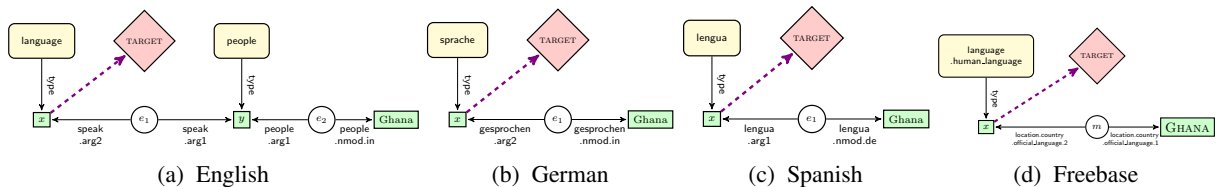

(a) English   (b) German   (c) Spanish   (d) Freebase

Figure 3: The ungrounded graphs for *What language do the people in Ghana speak?*, *Welche Sprache wird in Ghana gesprochen?* and *Cuál es la lengua de Ghana?*, and the corresponding grounded graph.

possible (for example, by treating `nsubjpass` as direct object). With the following entries,

*won* ∈ EVENT; *an*, *was* ∈ FUNCTIONAL; auxpass ∈ HEAD;
nsubjpass $= \lambda fgx. \exists y. f(x) \wedge g(y) \wedge \arg_2(x_e, y_a)$,

the lambda expression for *An Oscar was won* becomes $\lambda x. \text{won}(x_e) \wedge \text{Oscar}(y_a) \wedge \arg_2(x_e, y_a)$, identical to that of its active form. However, not having a special entry for passive verbs may have undesirable side-effects. For example, in the reduced-relative construction *Pixar claimed the Oscar won for Frozen*, the phrase *the Oscar won ...* will receive the semantics $\lambda x. \text{Oscar}(y_a) \wedge \text{won}(x_e) \wedge \mathbf{arg_1}(x_e, y_a)$, which differs from that of *an Oscar was won*. We leave it to the target application to disambiguate the interpretation in such cases.

**Long-Distance Dependencies** As discussed in Section 3.2, we handle long-distance dependencies evoked by clausal modifiers (`acl`) and control verbs (`xcomp`) with the BIND mechanism, whereas DEPLAMBDA cannot handle control constructions. For `xcomp`, as seen earlier, we use the mapping $\lambda fgx. \exists y. f(x) \wedge g(y) \wedge \text{xcomp}(x_e, y_e)$. For `acl` we use $\lambda fgx. \exists y. f(x) \wedge g(y)$, to conjoin the main clause and the modifier clause. However, not all `acl` clauses evoke long-distance dependencies, e.g. in *the news that Disney won an Oscar*, the clause *that Disney won an Oscar* is a subordinating conjunction of *news*. In such cases, we instead assign `acl` the INVERT semantics.

**Questions** Question words are marked with the enhanced part-of-speech tags DET:WH, ADV:WH and PRON:WH, which are all assigned the semantics $\lambda x. \${\text{word}}(x_a) \wedge \text{TARGET}(x_a)$. The predicate TARGET indicates that $x_a$ represents the variable of interest, that is the answer to the question.

### 3.4 Limitations

In order to achieve language independence, UDEP-LAMBDA has to sacrifice semantic specificity, since in many cases the semantics is carried by lexical information. Consider the sentences *John broke the*

*window* and *The window broke*. Although it is the *window* that broke in both cases, our inferred logical forms do not canonicalize the relation between *broke* and *window*. To achieve this, we would have to make the substitution of `nsubj` depend on lexical context, such that when *window* occurs as `nsubj` with *broke*, the predicate $\arg_2$ is invoked rather than $\arg_1$. We do not address this problem, and leave it to the target application to infer that $\arg_2$ and $\arg_1$ have the same semantic function in these cases. We anticipate that the ability to make such lexicalized semantic inferences in a task-agnostic cross-lingual framework would be highly useful and a crucial avenue for future work on universal semantics.

Other constructions that require lexical information are quantifiers like *every*, *some* and *most*, negation markers like *no* and *not*, and intentional verbs like *believe* and *said*. UD does not have special labels to indicate these. Although not currently implemented, we discuss how to handle quantifiers in this framework in the supplementary material.

## 4 Cross-lingual Semantic Parsing

To study the multilingual nature of UDEPLAMBDA, we conduct an empirical evaluation on question answering against Freebase in three different languages: English, Spanish, and German. Before discussing the details of this experiment, we briefly outline the semantic parsing framework employed.

### 4.1 Semantic Parsing as Graph Matching

UDEPLAMBDA generates *ungrounded* logical forms that are independent of any knowledge base, such as Freebase. We use GRAPHPARSER (Reddy et al., 2016) to map these logical forms to their grounded Freebase graphs, via corresponding ungrounded graphs. Figures 3(a) to 3(c) show the ungrounded graphs corresponding to logical forms from UDEPLAMBDA, each grounded to the same Freebase graph in Figure 3(d). Here, rectangles denote entities, circles denote events, rounded rectangles denote entity types, and edges between events

|  | WebQuestions |
|---|---|
| en | What language do the people in Ghana speak? |
| de | Welche Sprache wird in Ghana gesprochen? |
| es | ¿Cuál es la lengua de Ghana? |
| en | Who was Vincent van Gogh inspired by? |
| de | Von wem wurde Vincent van Gogh inspiriert? |
| es | ¿Qué inspiró a Van Gogh? |
|  | GraphQuestions |
| en | NASA has how many launch sites? |
| de | Wie viele Abschussbasen besitzt NASA? |
| es | ¿Cuántos sitios de despegue tiene NASA? |
| en | Which loudspeakers are heavier than 82.0 kg? |
| de | Welche Lautsprecher sind schwerer als 82.0 kg? |
| es | ¿Qué altavoces pesan más de 82.0 kg? |

Table 1: Example questions and their translations.

and entities denote predicates or Freebase relations. Finally, the TARGET node represents the set of values of $x$ that are consistent with the Freebase graph, that is the answer to the question.

GRAPHPARSER treats semantic parsing as a graph-matching problem with the goal of finding the Freebase graphs that are structurally isomorphic to an ungrounded graph and rank them according to a model. To account for structural mismatches, GRAPHPARSER uses two graph transformations: CONTRACT and EXPAND. In Figure 3(a) there are two edges between $x$ and *Ghana*. CONTRACT collapses one of these edges to create a graph isomorphic to Freebase. EXPAND, in contrast, adds edges to connect the graph in the case of disconnected components. The search space is explored by beam search and model parameters are estimated with the averaged structured perceptron (Collins, 2002) from training data consisting of question-answer pairs, using answer $F_1$-score as the objective.

## 4.2 Datasets

We evaluate our approach on two public benchmarks of question answering against Freebase: WebQuestions (Berant et al., 2013), a widely used benchmark consisting of English questions and their answers, and GraphQuestions (Su et al., 2016), a recently released dataset of English questions with both their answers and grounded logical forms. While WebQuestions is dominated by simple entity-attribute questions, GraphQuestions contains a large number of compositional questions involving aggregation (e.g. *How many children of Eddard Stark were born in Winterfell?*) and comparison (e.g. *In which month does the average rainfall of*

| $k$ | WebQuestions | | | GraphQuestions | | |
|---|---|---|---|---|---|---|
| | en | de | es | en | de | es |
| 1 | 89.6 | 82.8 | 86.7 | 47.2 | 39.9 | 39.5 |
| 10 | 95.7 | 91.2 | 94.0 | 56.9 | 48.4 | 51.6 |

Table 2: Structured perceptron $k$-best entity linking accuracies on the development sets.

*New York City exceed 86 mm?*). The number of training, development and test questions is 2644, 1134, and 2032, respectively, for WebQuestions and 1794, 764, and 2608 for GraphQuestions.

To support multilingual evaluation, we created translations of WebQuestions and GraphQuestions to German and Spanish.[6] For WebQuestions two professional annotators were hired per language, while for GraphQuestions we used a trusted pool of 20 annotators per language (with a single annotator per question). Examples of the original questions and their translations are provided in Table 1.

## 4.3 Implementation Details

Here we provide details on the syntactic analyzers employed, our entity resolution algorithm, and the features used by the grounding model.

**Dependency Parsing** The English, Spanish, and German Universal Dependencies (UD) treebanks (v1.3; Nivre et al 2016) were used to train part of speech taggers and dependency parsers. We used a bidirectional LSTM tagger (Plank et al., 2016) and a bidirectional LSTM shift-reduce parser (Kiperwasser and Goldberg, 2016). Both the tagger and parser require word embeddings. For English, we used GloVe embeddings (Pennington et al., 2014) trained on Wikipedia and the Gigaword corpus.[7] For German and Spanish, we used SENNA embeddings (Collobert et al., 2011; Al-Rfou et al., 2013) trained on Wikipedia corpora (589M words German; 397M words Spanish).[8] Measured on the UD test sets, the tagger accuracies are 94.5 (English), 92.2 (German), and 95.7 (Spanish), with corresponding labeled attachment parser scores of 81.8, 74.7, and 82.2.

**Entity Resolution** We follow Reddy et al. (2016) and resolve entities in three steps: (1) potential entity spans are identified using seven handcrafted part-of-speech patterns; (2) each span is associated

---

[6] Translations will be publicly released upon publication.
[7] http://nlp.stanford.edu/projects/glove/.
[8] https://sites.google.com/site/rmyeid/projects/polyglot.

| Method | WebQuestions | | | GraphQuestions | | |
|---|---|---|---|---|---|---|
| | en | de | es | en | de | es |
| SINGLEEVENT | 47.6 | 43.9 | 45.0 | 15.9 | 8.3 | 11.2 |
| DEPTREE | 47.8 | 43.9 | 44.5 | 15.8 | 7.9 | 11.0 |
| CCGGRAPH | 48.4 | – | – | 15.9 | – | – |
| UDEPLAMBDA | 48.3 | 44.2 | 45.7 | 17.6 | 9.0 | 12.4 |

Table 3: $F_1$-scores on the test for models trained on the training set (excluding the development set).

with potential Freebase entities according to the Freebase/KG API;[9] and (3) the 10-best entity linking lattices, scored by a structured perceptron, are input to GRAPHPARSER, leaving the final disambiguation to the semantic parsing problem. Table 2 shows the 1-best and 10-best entity disambiguation $F_1$-scores for each language and dataset.[10]

**Features** We use features similar to Reddy et al. (2016): *basic* features of words and Freebase relations, and *graph* features crossing ungrounded events with grounded relations, ungrounded types with grounded relations, and ungrounded answer type crossed with a binary feature indicating if the answer is a number. In addition, we add features encoding the *semantic* similarity of ungrounded events and Freebase relations. Specifically, we used the cosine similarity of the translation-invariant embeddings of Huang et al. (2015).[11]

### 4.4 Comparison Systems

We compared UDEPLAMBDA to prior work and three versions of GRAPHPARSER that operate on different representations: entity cliques, dependency trees, and CCG-based semantic derivations.

**SINGLEEVENT** This model resembles the learning-to-rank model of Bast and Haussmann (2015). An ungrounded graph is generated by connecting all entities in the question with the TARGET node, representing a single event. Note that this baseline cannot handle compositional questions, or those with aggregation or comparison.

**DEPTREE** An ungrounded graph is obtained directly from the original dependency tree. An event is created for each parent and its dependents in the tree. Each dependent is linked to this event with an edge labeled with its dependency relation, while the

---

[9]http://developers.google.com/freebase/.
[10]Due to the recent Freebase API shutdown, we used the KG API for GraphQuestions. We observed that this leads to inferior entity linking results compared to those of Freebase.
[11]http://128.2.220.95/multilingual/data/.

| Method | GraphQ. | WebQ. |
|---|---|---|
| SEMPRE (Berant et al., 2013) | 10.8 | 35.7 |
| JACANA (Yao and Van Durme, 2014) | 5.1 | 33.0 |
| PARASEMPRE (Berant and Liang, 2014) | 12.8 | 39.9 |
| QA (Yao, 2015) | – | 44.3 |
| AQQU (Bast and Haussmann, 2015) | – | 49.4 |
| AGENDAIL (Berant and Liang, 2015) | – | 49.7 |
| DEPLAMBDA (Reddy et al., 2016) | – | 50.3 |
| STAGG (Yih et al., 2015) | – | 48.4 (52.5) |
| BILSTM (Türe and Jojic, 2016) | – | 24.9 (52.2) |
| MCNN (Xu et al., 2016) | – | 47.0 (53.3) |
| AGENDAIL-RANK (Yavuz et al., 2016) | – | 51.6 (52.6) |
| UDEPLAMBDA | 17.6 | 49.5 |

Table 4: $F_1$-scores on the English GraphQuestions and WebQuestions test sets (results with additional task-specific resources in parentheses). Following prior work, for WebQuestions the union of the training and development sets were used for training.

parent is linked to the event with an edge labeled arg$_0$. If a word is a question word, an additional TARGET predicate is attached to its entity node.

**CCGGRAPH** This is the CCG-based semantic representation of Reddy et al. (2014). Note that this baseline exists only for English.

### 4.5 Results

Table 3 shows the performance of GRAPHPARSER with these different representations. Here and in what follows, we use average $F_1$-score of predicted answers (Berant et al., 2013) as the evaluation metric. We first observe that UDEPLAMBDA consistently outperforms the SINGLEEVENT and DEPTREE representations in all languages.[12]

For English, performance is almost on par with CCGGRAPH, which suggests that UDEPLAMBDA does not sacrifice too much specificity for universality. With both datasets, results are lower for German compared to Spanish. This agrees with the lower performance of the syntactic parser on the German portion of the UD treebank. Finally, while these results confirm that GraphQuestions is much harder compared to WebQuestions, we note that both datasets predominantly contain single-hop questions, as indicated by the competitive performance of SINGLEEVENT on both datasets.

Table 4 compares UDEPLAMBDA with previ-

---

[12]For the DEPTREE model, we CONTRACT each multi-hop path between the question word and an entity to a single edge. Without this constraint, DEPTREE $F_1$ results are 45.5 (en), 42.9 (de), and 44.2 (es) on WebQuestions, and 11.0 (en), 6.6 (de), and 2.6 (es) on GraphQuestions.

ously published models which exist only for English and have been mainly evaluated on WebQuestions. These are either symbolic like ours (first block) or employ neural networks (second block). Results for models using additional task-specific training resources, such as ClueWeb09, Wikipedia, or SimpleQuestions (Bordes et al., 2015) are shown in parentheses. On GraphQuestions, we achieve a new state-of-the-art result with a gain of 4.8 $F_1$-points over the previously reported best result. On WebQuestions we are 2.1 points below the best model using comparable resources, and 3.8 points below the state of the art. Most related to our work is the English-specific system of Reddy et al. (2016). We attribute the 0.8 point difference in $F_1$-score to their use of the more fine-grained PTB tag set and Stanford Dependencies.

## 5 Related Work

Our work continues the long tradition of building logical forms from syntactic representations initiated by Montague (1973). The literature is rife with attempts to develop semantic interfaces for HPSG (Copestake et al., 2005), LFG (Kaplan and Bresnan, 1982; Dalrymple et al., 1995; Crouch and King, 2006), TAG (Kallmeyer and Joshi, 2003; Gardent and Kallmeyer, 2003; Nesson and Shieber, 2006), and CCG (Steedman, 2000; Baldridge and Kruijff, 2002; Bos et al., 2004; Artzi et al., 2015). Unlike existing semantic interfaces, UDEPLAMBDA (like DEPLAMBDA) uses dependency syntax, taking advantage of recent advances in multilingual parsing (McDonald et al., 2013; Nivre et al, 2016).

A common trend in previous work on semantic interfaces is the reliance on rich typed feature structures or semantic types coupled with strong type constraints, which can be very informative but unavoidably language specific. Creating rich semantic types from dependency trees which lack a typing system would be labor intensive and brittle in the face of parsing errors. Instead, UDEPLAMBDA relies on generic unlexicalized information present in dependency treebanks and uses a simple type system (one type for dependency labels, and one for words) along with a combinatory mechanism, which avoids type collisions. Earlier attempts at extracting semantic representations from dependencies have mainly focused on language-specific dependency representations (Spreyer and Frank, 2005; Simov and Osenova, 2011; Hahn and Meurers, 2011; Reddy et al., 2016; Falke et al., 2016;

Beltagy, 2016), and multi-layered dependency annotations (Jakob et al., 2010; Bédaride and Gardent, 2011). In contrast, UDEPLAMBDA derives semantic representations for multiple languages in a common schema directly from Universal Dependencies. This work parallels a growing interest in creating other forms of multilingual semantic representations (Akbik et al., 2015; Vanderwende et al., 2015; White et al., 2016; Evang and Bos, 2016).

We evaluate UDEPLAMBDA on semantic parsing for question answering against a knowledge base. Here, the literature offers two main modeling paradigms: (1) learning of task-specific grammars that directly parse language to a grounded representation (Zelle and Mooney, 1996; Zettlemoyer and Collins, 2005; Wong and Mooney, 2007; Kwiatkowksi et al., 2010; Liang et al., 2011; Berant et al., 2013; Flanigan et al., 2014; Pasupat and Liang, 2015; Groschwitz et al., 2015); and (2) converting language to a linguistically motivated task-independent representation that is then mapped to a grounded representation (Kwiatkowski et al., 2013; Reddy et al., 2014; Krishnamurthy and Mitchell, 2015; Gardner and Krishnamurthy, 2017). Our work belongs to the latter paradigm, as we map natural language to Freebase indirectly via logical forms. Capitalizing on natural-language syntax affords interpretability, scalability, and reduced duplication of effort across applications (Bender et al., 2015). Our work also relates to literature on parsing multiple languages to a common executable representation (Cimiano et al., 2013; Haas and Riezler, 2016). However, existing approaches (Kwiatkowksi et al., 2010; Jones et al., 2012; Jie and Lu, 2014) still map to the target meaning representations (more or less) directly.

## 6 Conclusions

We introduced UDEPLAMBDA, a semantic interface for Universal Dependencies, and showed that the resulting semantic representation can be used for question-answering against a knowledge base in multiple languages. We provided translations of benchmark datasets in German and Spanish, in the hope to stimulate further multilingual research on semantic parsing and question answering in general. We have only scratched the surface when it comes to applying UDEPLAMBDA to natural language understanding tasks. In the future, we would like to explore how this framework can benefit other tasks such as summarization and machine translation.

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
