# Peer review of "Universal Semantic Parsing"

_ACL 2017 — decision unknown_

[Official Review · Reviewer 1 · rating 3 · confidence 4]
soundness 5 · originality 5 · clarity 4 · impact 3 · substance 5 · appropriateness 5 · meaningful comparison 3 · presentation format Oral Presentation

This paper describes interesting and ambitious work: the automated conversion
of Universal Dependency grammar structures into [what the paper calls] semantic
logical form representations.  In essence, each UD construct is assigned a
target construction in logical form, and a procedure is defined to effect the
conversion, working ‘inside-out’ using an intermediate form to ensure
proper nesting of substructures into encapsulating ones.  Two evaluations are
carried out: comparing the results to gold-standard lambda structures and
measuring the effectiveness of the resulting lambda expressions in actually
delivering the answers to questions from two QA sets.  

It is impossible to describe all this adequately in the space provided.  The
authors have taken some care to cover all principal parts, but there are still
many missing details.  I would love to see a longer version of the paper! 
Particularly the QA results are short-changed; it would have been nice to learn
which types of question are not handled, and which are not answered correctly,
and why not.  This information would have been useful to gaining better insight
into the limitations of the logical form representations.  

That leads to my main concern/objection.  This logical form representation is
not in fact a ‘real’ semantic one.                          It is, essentially, a
rather
close
rewrite of the dependency structure of the input, with some (good) steps toward
‘semanticization’, including the insertion of lambda operators, the
explicit inclusion of dropped arguments (via the enhancement operation), and
the introduction of appropriate types/units for such constructions as eventive
adjectives and nouns like “running horse” and “president in 2009”.  But
many (even simple) aspects of semantic are either not present (at least, not in
the paper) and/or simply wrong.  Missing: quantification (as in “every” or
“all”); numbers (as in “20” or “just over 1000”); various forms of
reference (as in “he”, “that man”, “what I said before”); negation
and modals, which change the semantics in interesting ways; inter-event
relationships (as in the subevent relationship between the events in “the
vacation was nice, but traveling was a pain”; etc. etc.  To add them one can
easily cheat, by treating these items as if they were just unusual words and
defining obvious and simple lambda formulas for them.  But they in fact require
specific treatment; for example, a number requires the creation of a separate
set object in the representation, with its own canonical variable (allowing
later text to refer to “one of them” and bind the variable properly).  For
another example, Person A’s model of an event may differ from Person B’s,
so one needs two representation symbols for the event, plus a coupling and
mapping between them.  For another example, one has to be able to handle time,
even if simply by temporally indexing events and states.  None of this is here,
and it is not immediately obvious how this would be added.  In some cases, as
DRT shows, quantifier and referential scoping is not trivial.  

It is easy to point to missing things, and unfair to the paper in some sense;
you can’t be expected to do it all.  But you cannot be allowed to make
obvious errors.  Very disturbing is the assignment of event relations strictly
in parallel with the verb’s (or noun’s) syntactic roles.  No-one can claim
seriously that “he broke the window” and “the window broke” has
“he” and “the window” filling the same semantic role for “break”. 
That’s simply not correct, and one cannot dismiss the problem, as the paper
does, to some nebulous subsequent semantic processing.                          This
really
needs
adequate treatment, even in this paper.  This is to my mind the principal
shortcoming of this work; for me this is the make-or-break point as to whether
I would fight to have the paper accepted in the conference.  (I would have been
far happier if the authors had simply acknowledged that this aspect is wrong
and will be worked on in future, with a sketch saying how: perhaps by reference
to FrameNet and semantic filler requirements.)                          

Independent of the representation, the notation conversion procedure is
reasonably clear.  I like the facts that it is rather cleaner and simpler than
its predecessor (based on Stanford dependencies), and also that the authors
have the courage of submitting non-neural work to the ACL in these days of
unbridled and giddy enthusiasm for anything neural.